# DexUMI: Using Human Hand as the Universal Manipulation Interface for Dexterous Manipulation

**Abstract:** We present DexUMI - a data collection and policy learning framework that uses the human hand as the natural interface to transfer dexterous manipulation skills to various robot hands. DexUMI includes hardware and software adaptations to minimize the embodiment gap between the human hand and various robot hands. The hardware adaptation bridges the kinematics gap using a wearable hand exoskeleton. It allows direct haptic feedback in manipulation data collection and adapts human motion to feasible robot hand motion. The software adaptation bridges the visual gap by replacing the human hand in video data with high-fidelity robot hand inpainting. We demonstrate DexUMI's capabilities through comprehensive real-world experiments on two different dexterous robot hand hardware platforms, achieving an average task success rate of 86%.

**Keywords:** Dexterous Manipulation, Learning from Human, Imitation Learning

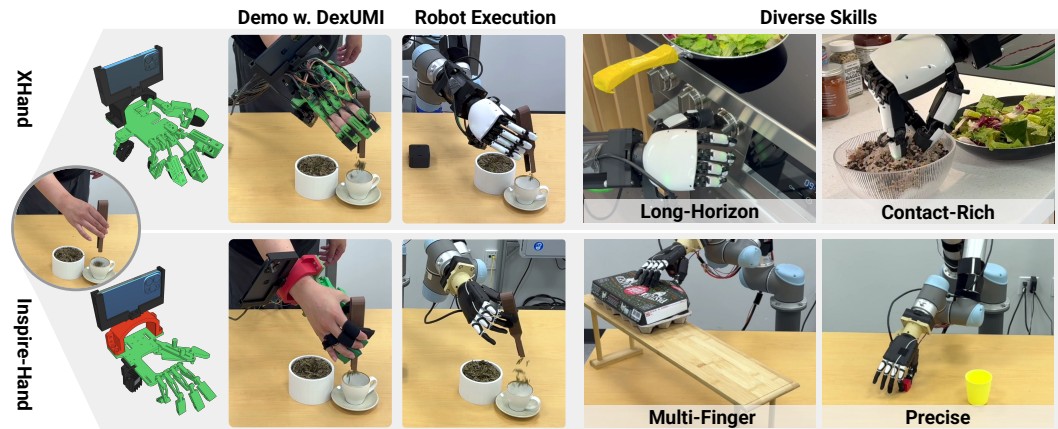

Figure 1: **DexUMI** transfer dexterous human manipulation skills to various robot hand by using wearable exoskeletons and a data processing framework. We demonstrate DexUMI's capability and effectiveness on both underactuated (e.g., Inspire) and fully-actuated (e.g., XHand) robot hand for a wide variety of manipulation tasks.

## 1 Introduction

Human hands are incredibly dexterous in a wide range of tasks. Dexterous robot hands are designed with the hope of replicating this capability. However, it remains a significant challenge to transfer skills from human hands to robotic counterparts due to their substantial *embodiment gap*. This gap manifests in various forms, such as differences in kinematic structures, contact surface shape, available tactile information, and visual appearance.

---

∗ Indicates equal contribution

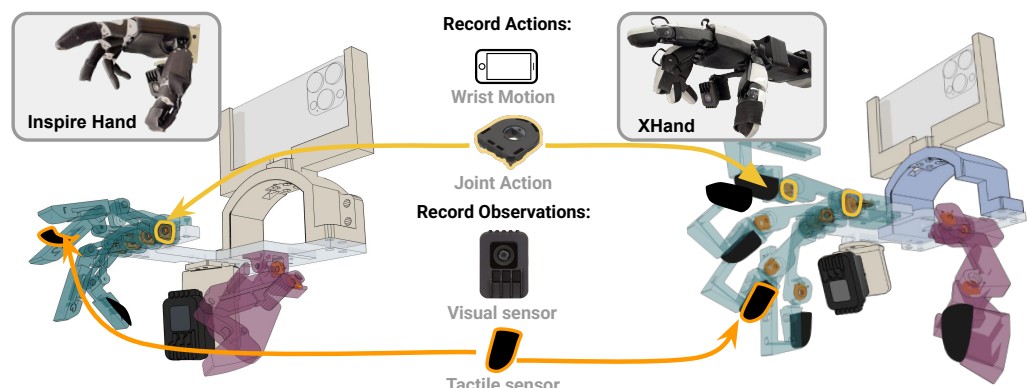

Figure 2: **Exoskeleton Design.** The optimized exoskeleton design shares the same joint-to-fingertip position mapping as the target robot hand while maintaining the wearability. The exoskeletons utilizes the encoder to precisely capture the joint action and 150° DFoV camera to record the information-rich visual observation. An iPhone is rigidly mounted to track the wrist pose through the ARKit.

Teleoperation has become a popular manipulation interface for dexterous hands. However, teleoperation can be difficult due to the spatial observation mismatch and the lack of direct haptic feedback. These problems do not exist when human hand can perform the manipulation task directly. In other words, human hand itself is a better manipulation interface. In this paper, we ask the following question:How can we minimize the embodiment gap, so that we can use the human hand as the universal manipulation interface for diverse robot hands?

To answer this question, we propose **DexUMI**, a framework with hardware and software adaptation components that is designed to minimize the action and observation gaps. The **hardware adaptation** takes the form of a wearable hand exoskeleton. A user can directly collect manipulation data while wearing it. The exoskeleton is designed for each target robot hand through *a hardware optimization framework* that refines exoskeleton parameters (e.g., link lengths) to closely match the robot finger trajectories while maintaining wearability for the human hand. Our **software adaptation** takes the form of a data processing pipeline that bridges the visual observation gap between human demonstration and robot deployment.

With both hardware and software adaptation layers, DexUMI allows us to collect data on various tasks with minimal kinematic and visual gaps then transfer skills to robots. Comprehensive real-world experiments demonstrate DexUMI's capability on two different dexterous hand types: a 6-DoF Inspire hand [1] and a 12-DoF XHand [2]. Our approach achieves 3.2 times greater data collection efficiency compared to teleoperation and an average success rate of 86% across four tasks , including long-horizon and complex tasks requiring multi-finger contacts.

## 2 Hardware Adaptation to Bridge the Embodiment Gap

### 2.1 Exoskeleton Mechanism Design

Modern robot hands often closely mimic human hands anatomically, meaning that a hand exoskeleton would compete for space with the human hand wearing it. Our exoskeleton design has two goals to achieve:

1. *Shared joint-action mapping:* The exoskeleton and the target robot hand must share the same joint-to-fingertip position mapping, including their limits, so the action can transfer.
2. *Wearability:* The exoskeleton must allow sufficient natural movements of the user's hand.

While the first goal can be mathematically defined, the wearability goal is hard to write down concretely. Our solution is to parameterize the exoskeleton design and formulate the wearability requirements as constraints on the design parameters, then find a solution that accommodates wearability while preserving kinematic relationships by solving an optimization.

***E.1 Design initialization:*** We initialize the design with parameterized robot hand models based on URDF files (See Fig. 3). When such detailed designs are unavailable (e.g., the Inspire-Hand's finger

54 mechanisms), we substitute them with equivalent general linkage designs with the same DoFs (e.g.,
55 a four-bar linkage) and allow optimization to find parameters that best match the observed kinematic
56 behavior. Please see Appendix for details.

57 **E.2 Bi-level optimization objective:** Our
58 optimization objective maximizes the fol-
59 lowing similarity: $\max_{\mathbf{p}} \mathcal{S}(\mathcal{W}_{\text{exo}}^{\text{tip}}(\mathbf{p}), \mathcal{W}_{\text{robot}}^{\text{tip}})$,
60 where $\mathcal{W}_{\text{exo}}^{\text{tip}}$ and $\mathcal{W}_{\text{robot}}^{\text{tip}}$ represent the finger-
61 tip workspaces (set of all possible fingertip
62 pose in SE(3)) for the exoskeleton and robot
63 hand, respectively. $\mathbf{p} = \{j_1, ..., j_n, l_1, ..., l_m\}$
64 is the exoskeleton design parameters including
65 joint positions $j_i \in \mathbb{R}^3$ in the wrist coordi-
66 nate (i.e., flange) and linkage lengths $l_j$. The
67 function $\mathcal{S}(\cdot, \cdot)$ represents a similarity metric
68 between the two workspaces, which quantifies

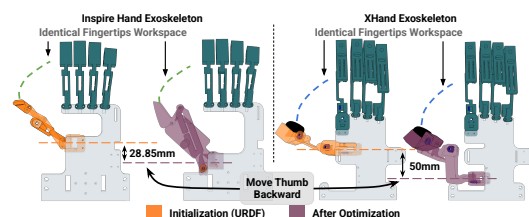

Figure 3: **Mechanism Optimization.** To avoid thumb collision between human hand and exoskeleton, the hardware optimization step allows us to move the exoskeleton thumb backward while still preserving the original fingertip and joint mapping in SE(3) space.

69 how closely the exoskeleton's fingertip pose distribution matches that of the robot hand.

70 **E.3 Constraints**: We apply bound constraints $j_i \in \mathcal{C}_i$ and $l_j^{\min} \leq l_j \leq l_j^{\max}$, which are empirically
71 selected to ensure that the exoskeleton can be comfortably worn.

## 2.2 Sensor Integration

73 Sensors on the exoskeleton need to satisfy the following design objectives: *1. Capture sufficient*
74 *information:* the sensors need to capture ALL the information necessary for policy learning, which
75 includes: robot action such as joint angle (*S.1*) and wrist motion (*S.2*), as well as observations in
76 both vision (*S.3*) and tactile (*S.4*). *2. Minimize embodiment gap:* the sensory information should
77 have minimal distribution shift between human demonstration and robot deployment.

78 **S.1 Joint capture & mapping.** To precisely capture joint actions, our exoskeleton integrates joint
79 encoders at every *actuated* joint – using resistive position encoders for both the XHand and Inspire-
80 hand. We choose the `Alps` encoder [3] for its size and precision. **S.2 Wrist pose tracking.** We use
81 iPhone ARKit to capture the 6DoF wrist pose, as smartphones represent the most accessible devices
82 capable of providing precise spatial tracking. **S.3 Visual observation.** We mounted a 150° diagonal
83 field of view (DFoV) wide-angle camera `OAK-1` [4] under the wrist for both the exoskeleton and
84 the target robot dexterous hand. **S.4 Tactile sensing.** We install tactile sensors on the exoskeleton to
85 capture and translate these tactile interactions. To ensure consistent sensor readings, we install the
86 same type of tactile sensors on the exoskeleton as those used on the target robot hand.

## 3 Software Adaptation to Bridge the Visual Gap

88 Fig. 4 shows the visual gap between human
89 demonstration (a) and robot deployment (h).
90 The adaptation takes four steps: *V.1 Segment*
91 *human hand and exoskeleton.* Firstly, we seg-
92 ment (Fig. 4b) the human hand and exoskele-
93 ton on observation videos using SAM2 [5]. *V.2*
94 *Inpaint environment background.* With seg-
95 mentation, we remove the human hand and the
96 exoskeleton pixels from the image data. Then

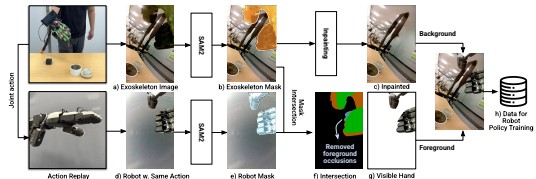

Figure 4: **Bridging the Visual Gap.** To convert the visual observation into policy training data.

97 we use ProPainter [6], a flow-based inpainting method, to fully refill (Fig. 4c) the missing areas [7–
98 9]. *V.3 Record corresponding robot hand video.* Next, to render robot hand properly into the video,
99 we replay the recorded joint action on the robot hand and record another video with only the robot
100 hand (Fig. 4d). This step does not involve the robot arm. We then used SAM2 again to extract the
101 robot hand pixels (Fig. 4e) and discard the background. *V.4 Compose robot demonstrations.* The
102 last step is to merge the inpainted-background-only video with robot-hand-only video. It is crucial
103 to maintain occlusion relationships: the robot hand does not always appear on top. We developed an
104 occlusion-aware compositing approach leveraging: (1) our consistent under-wrist camera setup and

| Method | | | Inspire Hand | | | | XHand | | | | |
|--------|--------|--------|------|--------|------|------|------|------|------|------|------|
| | | | | | Tea | | Tea | | Kitchen | | |
| Action | Tactile | Visual | Cube | Carton | tool | leaf | tool | leaf | knob | pan | salt |
| Rel | Yes | Inpaint | **1.00** | 0.85 | **1.00** | 0.85 | **1.00** | **0.85** | **0.95** | **0.95** | **0.75** |
| Abs | Yes | Inpaint | 0.10 | 0.35 | 0.80 | 0.00 | **1.00** | 0.25 | 0.50 | 0.45 | 0.00 |
| Rel | No | Inpaint | 0.95 | **0.90** | **1.00** | **0.90** | 0.95 | 0.80 | **0.95** | **0.95** | 0.15 |
| Abs | No | Inpaint | 0.90 | 0.85 | 0.90 | 0.60 | **1.00** | 0.75 | 0.60 | 0.60 | 0.0 |
| Rel | No | Mask | 0.60 | 0.10 | 0.90 | 0.50 | / | / | / | / | / |
| Rel | No | Raw | 0.20 | 0.05 | 0.85 | 0.05 | / | / | / | / | / |

Table 1: **Evaluation Results.** We report stage-wise accumulated success rate. The experiments compare different combinations of finger action representation (Absolute vs Relative), tactile feedback (Yes vs No), and visual rendering approaches (Inpaint vs Mask/Raw).

(2) the kinematic and shape similarity between the exoskeleton and robot hand. We compute a visible mask (Fig. 4f) by intersecting the exoskeleton mask and robot hand mask. Rather than naively overwriting pixels, we selectively replace pixels in the inpainted observation with robot hand pixels only if those pixels are present in the visible mask.

## 4 Evaluation

**DexUMI framework enables efficient dexterous policy learning:** As shown in Tab. 1, the DexUMI system achieves high success rates across all four tasks on two robot hands. The system handles precise manipulation, long-horizon tasks, and coordinated multi-finger contact, while effectively generalizing across diverse manipulation scenarios.

**Relative finger trajectories are more robust to noise and hardware imperfections:** Tab. 1 shows relative finger trajectory consistently achieves better success across all tasks. Fig. 5 shows more insights: relative trajectory can make critical contact events more reliable.

**Tactile feedback improves performance on tasks with clean force profiles:** We focused on the XHand as its tactile sensors provide cleaner readings. We observed that tactile feedback significantly improved performance on picking up salt. This task highlights the effect of tactile because 1) The tactile sensors give a clear, large reading when the fingers touch the bowl of salt. 2) There is little useful visual information close to grasping as the camera view is mostly blocked by the bowl.

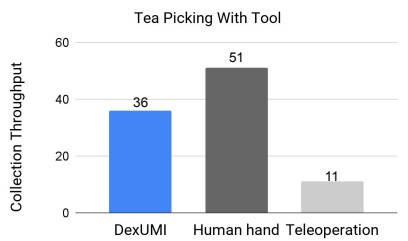

Figure 5: **Comparisons.** a) The policy outputs relative hand actions yield more precise action and demonstrate better multi-finger coordination. b) Even with noisy tactile sensor reading, the tactile significantly improve tasks which is visually challenging.

Figure 6: **Efficiency:** Collection throughput (CT) within 15-minute.

**DexUMI framework enables efficient dexterous hand data collection:** We compared data collection efficiency across three ways: DexUMI, bare human hand, and teleoperation on the tea-picking-with-tool task. As illustrated in Fig. 6, while DexUMI remains slower than direct human hand manipulation, it achieves 3.2 times greater efficiency than traditional teleoperation methods.

## 5 Conclusion

We present DexUMI, a scalable and efficient data collection and policy learning framework that uses the human hand as an interface to transfer human hand motion to precise robot hand actions while providing natural haptic feedback. Our work establishes a new approach to collecting real-world dexterous hand data efficiently and at scale beyond traditional teleoperation.

# 6 Limitation and Future Work

We would like to discuss DexUMI's limitations from three different aspects: hardware adaptation, software adaptation, and existing robot hand hardware.

**Hardware Adaptation:**

- *Per robot hand exoskeleton design:* Although DexUMI demonstrates generalizability across underactuated and fully-actuated hands, our optimization framework still requires hardware-specific tuning, especially for wearability. One future work direction is fully automated optimization formulation given robot hand model and some description of the human hand. Further, our hardware optimization framework can potentially leverage generative models [10] to increase efficiency and accuracy when design space grows.

- *Fingertips Matching:* Our current formulation focuses only on matching the fingertip workspace between the designed exoskeleton and target robot hand. It would be interesting for future work to also model remaining potential contact geometries such as the palm.

- *Wearability:* The hardware optimization pipeline makes the exoskeleton wearable and allows humans to operate it relatively easily for extended periods. However, wearability could be further improved by integrating soft materials, such as TPU for parts that contact the human hand. Additionally, constrained by both the design of the target hand and 3D printing material strength, users might still experience limitations in fully stretching certain fingers.

- *Reliability of Tactile Sensors:* Throughout our experiments, we found that reliable tactile sensors are key to maintaining consistent tactile observation between the exoskeleton and corresponding robot hand, thereby reducing the embodiment gap. In our implementation, the resistive tactile sensors added to the Inspire hand and its exoskeleton proved sensitive to their attachment way on fingers. Meanwhile, the electromagnetic tactile sensors on the XHand and its exoskeleton showed a tendency to drift after exposure to high pressure. Since the human hand generates more force than the robot hand, tactile sensor readings frequently drift when humans operate the exoskeleton. Future work can also incorporate other types of tactile sensors, such as vision-based tactile sensors [11–13] and capacitive F/T sensors [14].

- *Material Limitations:* Our experiments demonstrate that DexUMI is able to capture fine-grained fingertip actions such as closing tweezers. However, we sometimes found that encoders cannot precisely capture human motion due to 3D printing material strength limitations; occasionally, the human hand slightly distorts the exoskeleton linkage when manipulating objects. In such cases, encoders are unable to capture this distortion.

**Software Adaptation:**

- *Robot Hand Image:* Currently, we still require real-world robot hardware to obtain robot hand images. However, this requirement could be eliminated by implementing an image generation model that receives motor values as input and produces corresponding hand pose images as output.

- *Inpainting Quality:* Throughout our experiments, we found that the current software adaptation pipeline can already yield high-fidelity robot hand images. Nevertheless, we observed that illumination effects on the robot hand cannot be fully reproduced, and some areas in the image appear blurred due to limitations in the inpainting process.

- *Camera Location:* DexUMI currently requires the camera to be rigidly attached to the robot hand/exoskeleton and does not support a moving camera. However, it would be feasible to collect a dataset and train an image generation model that receives the relative pose between the camera and hand, along with hand pose information, to generate the corresponding hand pose image from any given camera position.

**Existing Robot Hand Hardware:**

- *Precision:* Throughout our experiments, we found that both the Inspire Hand and XHand lack sufficient precision due to backlash and friction. For example, the fingertip location of the Inspire Hand differs when moving from 1000 to 500 motor units compared to moving from 0 to 500

motor units. Although the desired motor value is the same in both cases, the final fingertip position varies. We observed this phenomenon in both robot hands. Consequently, when fitting regression models between encoder and hand motor values, we can typically ensure precision in only "one direction"—either when closing the hand or opening it. This inevitably causes minor discrepancies in the inpainting and action mapping processes. Further, we found that the XHand mapping between motor command and fingertip location slightly differs across time shifts or after each reboot.

- *Size Discrepancy:* The size difference between the robot hand and the human hand may cause wearability issues. For example, if the robot hand is twice as large as the human hand, it becomes difficult for both the human hand and the exoskeleton to reach the joint configurations required by the robot hand.
- *Co-design:* Many of these wearability issues arise from design constraints in existing commercial hardware. An interesting direction would be to explore a reverse design paradigm: first designing an exoskeleton that is comfortable and fully operable for humans, and then using that exoskeleton as the foundation for designing the robot hand.

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

# Appendix

# A   Additional Experiment Results

## A.1   Inpainting Results

We show the processed visual observation by the software adaptation layer in policy training data in Fig. 7. Our software adaptation bridges the visual gap by replacing the human hand and exoskeleton in visual observations recorded by the wrist camera with high-fidelity robot hand inpainting. Though the overall inpainting quality is good, we found there are still some deficiencies in the output caused by:

- **Imperfect Segmentation from SAM2:** In most cases, SAM2 can segment the human hand and exoskeleton effectively. However, we notice SAM2 sometimes misses some small areas on the exoskeleton.
- **Quality of inpainting method:** We use flow-based inpainting to replace the human and exoskeleton pixels with background pixels. Though the overall quality is high, some areas remain blurry. We add Gaussian blur augmentation to the images during policy training to make the policy less sensitive to this blurriness.
- **Robot hand hardware limitations:** Throughout our experiments, we found that both the Inspire Hand and XHand lack sufficient precision due to backlash and friction. For example, the fingertip location of the Inspire Hand differs when moving from 1000 to 500 motor units compared to moving from 0 to 500 motor units. Consequently, when fitting regression models between encoder and hand motor values, we can typically ensure precision in only "one direction"—either when closing the hand or opening it. This inevitably causes minor discrepancies in the inpainting and action mapping processes.
- **Inconsistent illumination:** Similar to prior work [9], we found that illumination on the robot hand might be inconsistent with what the robot experiences during deployment. Therefore, we add image augmentation including color jitter and random grayscale during policy training to make the learned policy less sensitive to lighting conditions.
- **3D-printed exoskeleton deformation:** The human hand is powerful and can sometimes cause the 3D-printed exoskeleton to deform during operation. In such cases, the encoder value fails to reflect this deformation. Consequently, the robot finger location might not align with the exoskeleton's actual finger position.

## A.2   Relative and Absolute Action Distribution

We visualize both relative and absolute action distribution of thumb swing joints in the Kitchen task. The relative action distribution is a simple unimodal while the absolute action distribution is multi-modal.

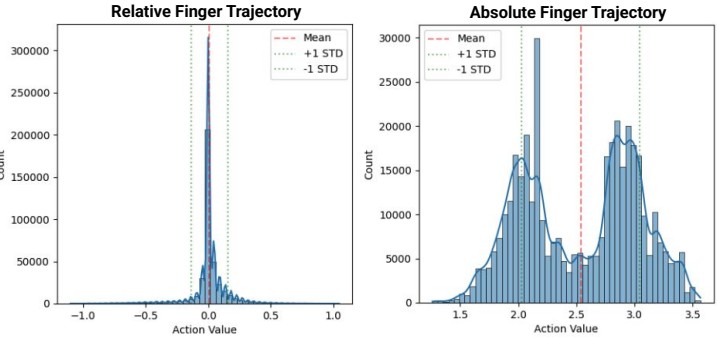

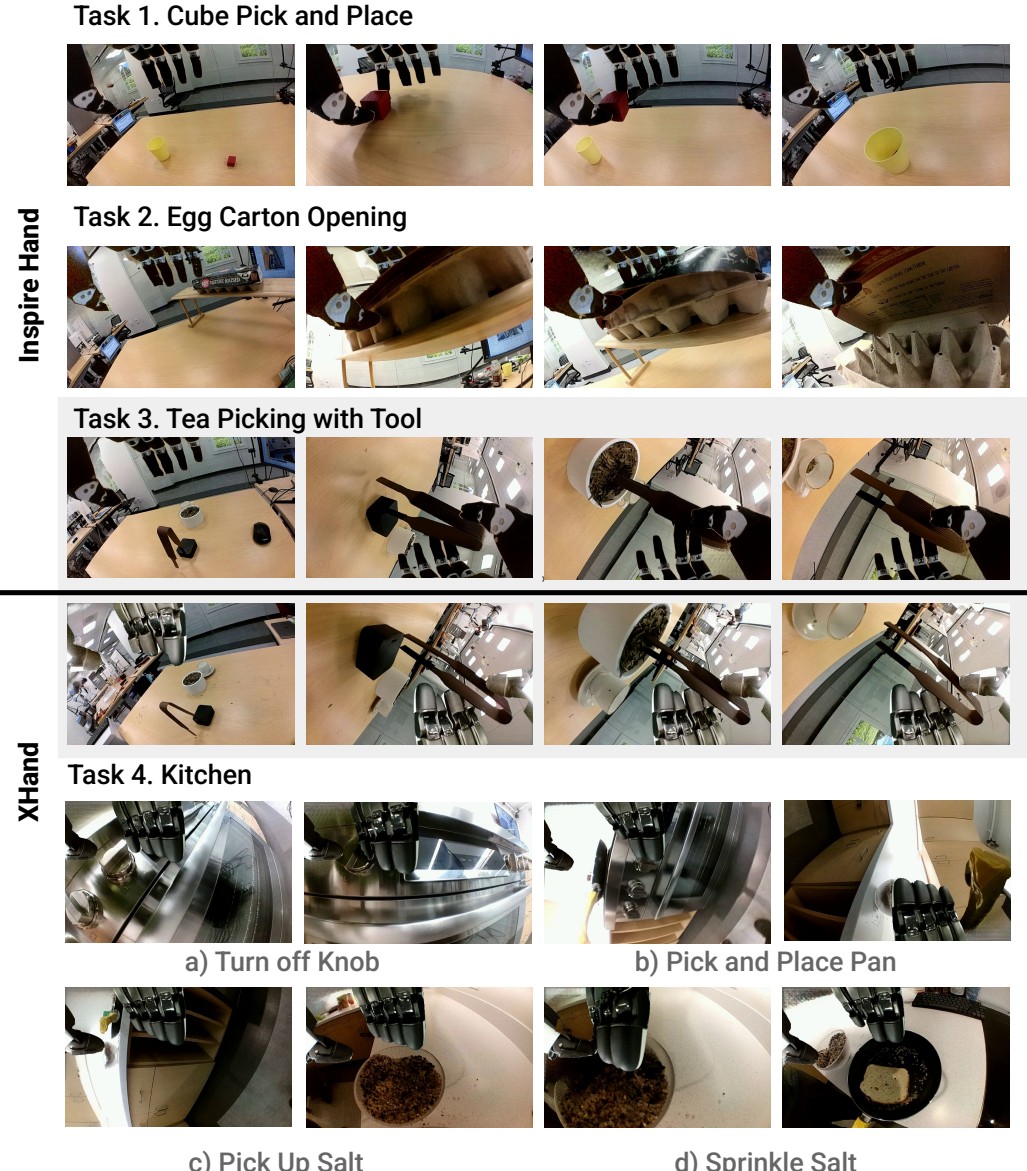

**Task 1. Cube Pick and Place**

**Inspire Hand**

**Task 2. Egg Carton Opening**

**Task 3. Tea Picking with Tool**

**XHand**

**Task 4. Kitchen**

a) Turn off Knob          b) Pick and Place Pan

c) Pick Up Salt          d) Sprinkle Salt

Figure 7: **Inpainting Results.** The visual observations in the original collected dataset contain exoskeletons and human hands. The software adaptation layer replaces these pixels with corresponding robot hand images while preserving the natural occlusion relationships during hand-object interactions. Please see project website https://dex-umi.github.io for details.

# B  Evaluation Details

## B.1  Initial State Selection

For each task, we manually select a set of initial states for the environment. Objects are placed as diversely as possible within the environment. This set of initial states is shared across all methods. We achieve consistency by placing an additional side camera to record images of all selected initial states. When starting a new evaluation episode, we visualize an image overlay between the recorded pre-selected initial state and the current initial state. We carefully adjust the current setup until it matches the pre-selected initial state with near pixel-perfect alignment.

Note that due to differences in wrist camera placement relative to the robot flange between the XHand and Inspire Hand, some initial states viable for the Inspire Hand cannot be completed by the XHand. For example, if the tea cup is positioned more than $45°$ to the left of the tea pot (image space), the XHand's wrist camera cannot capture the tea cup after grasping the tea due to its camera positioning (the XHand thumb has a larger range of motion, requiring us to rotate the wrist camera more toward the thumb direction to obtain clearer visual observations). Consequently, the XHand and Inspire Hand do not strictly share the same set of initial states for the Tea Picking Using Tool task. Nevertheless, we ensure their initial states remain within similar distributions and maintain as much diversity as possible.

For the kitchen task, the large workspace presents challenges for a fixed-base single UR5 to cover diverse initial states, particularly regarding the seasoning bowl location, as the stove and knob positions are fixed. Despite these constraints, we maximize the diversity of bowl placement within the kinematically feasible workspace.

## B.2   Success Criteria

**Cube Picking:** The robot must pick up the red cube and place it into the yellow cup. If the cup falls over after the cube is already placed in it, we still count the episode as successful.

**Egg Carton:** We define task success as when the lid is lifted up with its box at an angle greater than $30°$ and the egg box remains stable on the shelf.

**Tea Picking Using Tool:** This task consists of two sub-tasks. We define tool picking success as the robot's ability to steadily hold the tweezers and move them to the tea pot. We define leaf picking success as the robot's ability to use tweezers to 1) grasp at least one tea leaf from the pot and 2) transfer at least half of the grasped tea leaf into the cup. Subsequent sub-tasks automatically count as failures if the previous sub-task fails, even if the robot can successfully complete the later sub-tasks.

**Kitchen Manipulation:** This task consists of three sub-tasks. We define knob closing success as the robot hand rotating the knob by at least $60°$ from its initial position. We define pan moving success as the robot moving the pan from the stove to the counter without dropping it during transfer. We define the salt task success as the robot 1) grasping some seasoning from the bowl and 2) sprinkling it inside the pan. Subsequent sub-tasks automatically count as failures if the previous sub-task fails, even if the robot can successfully complete the later sub-tasks.

## B.3   Policy Execution

The learned policy predicts 16 steps of future actions, but the robot only executes the first 8 steps and discards the rest. The policy executes at 10 Hz, while the UR5 executes commands at 125 Hz. The Inspire Hand executes at 10 Hz, and the XHand executes at 60 Hz. The 10 Hz policy commands are linearly interpolated to match the desired hardware execution frequency.

The action output by the policy contains two components: relative UR5 end-effector action and hand action. The relative end-effector action from the learned policy is converted to absolute by adding the relative action to the current UR5 absolute position in the UR5 base frame. For hand actions, if the action type is absolute, the desired motor value is sent directly to the robot hand for execution. If the hand action type is relative, we first read the current hand motor position, add the relative hand action to it, and then send the result for execution.

For the XHand, we found that creating a virtual current hand motor position improves performance compared to reading the current position directly from hardware. Unlike the Inspire Hand motor, which is self-locking, the XHand finger position slightly drifts after encountering external forces (such as the restoring force of tweezers). The 10 Hz policy isn't reactive enough to adjust for this real-time drifting. Consider the following scenario: the robot hand attempts to close the tweezers to grasp tea leaves. The current motor value obtained by calling the hardware API might already be outdated due to the restoring force of the tweezers (causing fingers to spread wider) when robot

execution begins. To address this issue, we initialize a virtual current hand motor position by reading the actual motor position at the beginning of the evaluation. Once the evaluation begins, we update this virtual hand motor position by adding the executed relative hand actions. With this virtual hand motor position approach, finger actions become less impacted by physical drifting, resulting in more precise and reliable grasping operations.

## C    Exoskeleton Design Details

### C.1    Inspire Hand

Underactuated hands like the Inspire Hand typically incorporate closed-loop kinematics, such as four-bar linkages, which cannot be directly represented in URDF. As a result, we cannot initialize the exoskeleton design for the Inspire Hand directly from its URDF model. Instead, our approach is to capture the finger kinematic behavior—specifically, the fingertip poses—and use equivalent general linkage designs with the same degrees of freedom (DoFs) as an initial template for the finger mechanisms. This allows the optimization process to identify parameters that best match the observed kinematics.

To achieve this, we employed a motion capture system (see Fig. 8) to record the fingertip poses in SE(3) space. We 3D-printed marker mounting components for each finger and flange and installed them on the Inspire Hand. For the index, middle, ring, and pinky fingers, each of which has a single DoF, we uniformly sampled 16 motor command values from the lower limit (0) to the upper limit (1000), sent the commands to the fingers, and recorded the corresponding fingertip poses.

For the thumb, which has 2 DoFs—swing and bend—we first fixed the swing value and then uniformly sampled the bend motor values. For example, as shown in Fig. 8d, we set the swing motor to 400 and recorded the fingertip poses by varying the bend motor command. We repeated this procedure for swing values of 0, 200, 400, 600, 800, and 1000.

After obtaining the fingertip poses in the flange coordinate system, we applied the same bi-level optimization formulation defined in Equation 1 in main paper to determine design parameters for each finger. For all five fingers, we employed four-bar linkages as the linkage designs. For each sampled design parameter, We simulate the fingertip poses using PlaCo [15]. For thumb, we minimized the overall loss across all swing motor values, since the thumb's structural configuration should remain consistent regardless of the swing motor value.

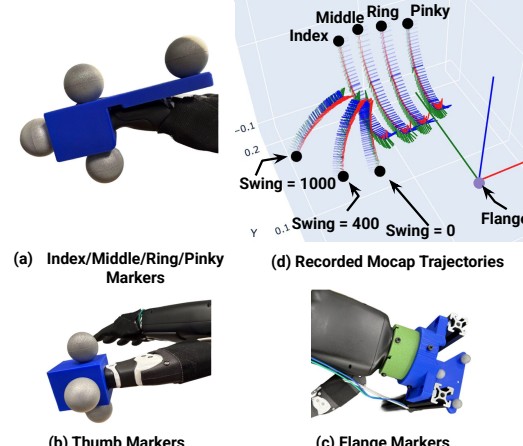

(a) Index/Middle/Ring/Pinky Markers

(d) Recorded Mocap Trajectories

(b) Thumb Markers

(c) Flange Markers

Figure 8: **Inspire Mocap:** We use motion capture system to record fingertips trajectories in the flange coordinate. We attached marker on fingers and flange to capture the fingertip pose in flange coordinate.

From the optimized design parameters to the physical implementation, we apply three additional steps to ensure that the exoskeleton mask consistently covers the real Inspire Hand. First, we extend the length of the last link of each finger in the exoskeleton design by 3 mm beyond the optimized value. This guarantees that the exoskeleton mask always fully covers the last link of the actual Inspire Hand. Second, we increase the width of the thumb's four-bar linkage to eliminate any hollow regions in the camera's field of view, thereby maintaining the visual integrity of a continuous exoskeleton mask. Third, we conservatively tighten the joint limits by 5° at each joint to ensure the mask continues to cover the real Inspire Hand even when structural deformation occurs due to the limited strength of the 3D-printed PLA-CF material.

## C.2  XHand

Since the URDF file of the XHand is well-organized, with each joint origin defined at the location of its corresponding rotary joint, we can directly extract link lengths from the URDF structure. In cases where the exact values are not specified, we can perform reverse modeling using the STL meshes from URDF file to recover geometric features near each joint and manually measure link lengths in CAD software.

Joint limits are also specified in the URDF file and are implemented in the exoskeleton design by physically constraining the link motion to prevent rotation beyond the specified range. Similar to the Inspire Hand exoskeleton design, we adopt a conservative strategy when applying these limits setting slightly tighter bounds on each joints. For example, if the actual joint rotation range is $-110°$ to $20°$, the corresponding exoskeleton limit is set to $-105°$ to $15°$. This precaution accounts for possible deformation of the 3D-printed exoskeleton links under human-applied torque, which can introduce unintended joint deflection. Without this buffer, the exoskeleton might deform beyond the physical limits of the XHand, leading to an embodiment gap.

When converting the link lengths to the actual exoskeleton design, two primary constraints must be considered. The first is *wearability*. To ensure that the human operator can comfortably wear the exoskeleton, the structure must be hollowed out as much as possible, allowing the finger to pass through unobstructed. The second constraint is *material strength*. Through empirical testing, we determined that the optimal minimum structural width for 3D-printed PLA-CF material is 4 mm. Therefore, any part expected to experience significant stress is reinforced to be at least 4 mm thick in the final design.

# D  Sensor Details

## D.1  Joint Encoder

Our exoskeleton uses Alps RDC506018A rotary sensors as encoders at every joints. These are resistive sensors whose resistance varies approximately linearly with absolute angular position.

As shown in Fig. 9, when the joint rotates, the voltage on the ADC line changes proportionally. This analog voltage signal is then sampled by an Analog-to-Digital Converter (ADC) on a microcontroller unit (MCU). Then the joint angle $\alpha_{\text{joint}}$ can be estimated as:

$$\alpha_{\text{joint}} = \frac{V_{\text{ADC}}}{3.3\,\text{V}} \times 360°$$

Figure 9: **Joint Encoder Circuit:** The rotary sensor acts as a variable resistor with three output pins. As it rotates with the joint, the voltage on the ADC line changes approximately linearly.

However, this simple voltage divider circuit has a significant failure mode: if the power supply (3.3 V in our case) is unstable due to temperature drift in semiconductor components or ripple from DC-DC converters and LDOs, the joint angle reading will drift accordingly. To mitigate this issue, we simultaneously measure the supply voltage through another ADC channel. Instead of dividing by a fixed 3.3 V, we normalize the sensor voltage using the measured supply voltage when computing the joint angle:

$$\alpha_{\text{joint}} = \frac{V_{\text{ADC}}}{V_{\text{supply}}} \times 360°$$

This voltage normalization runs in real time on the MCU. After computing the joint angles, the MCU packs all joint values into a single data packet with a fixed 2-byte header and a checksum tail. The header simplifies decoding by allowing the receiver to locate a known keyword in variable-length data streams, while the checksum ensures packet integrity. The final data packet is transmitted to the host computer via a Universal Asynchronous Receiver-Transmitter (UART) interface.

## D.2 Tactile

For commercial dexterous hands without built-in tactile sensors (e.g., the Inspire Hand in our evaluation), we use a simple and low-cost Force-Sensitive Resistor (FSR) as the tactile sensor. When no force is applied, the FSR exhibits a resistance of several megaohms, while under significant force, the resistance drops to the kiloohms range. As shown in Fig. 10, the FSR is incorporated into a simple voltage divider circuit to produce an analog voltage signal. The divider resistor $R_1$ is selected to be comparable to the minimum resistance of the FSR. Since the FSR resistance is approximately inversely proportional to the applied force, we can express the force using a constant scale factor $k$ as:

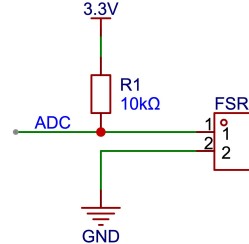

Figure 10: **Voltage Divider Circuit:** This simple voltage divider circuit converts the resistance change of the FSR sensor into an analog voltage on the ADC line.

$$F = k \left( \frac{V_{\text{supply}}}{V_{\text{ADC}}} - 1 \right)$$

In our experimental setup, the same FSR sensor is mounted on both the dexterous hand and the exoskeleton. For simplicity, we directly use the $V_{\text{ADC}}$ reading as a proxy for tactile input.

For hands equipped with onboard tactile sensors (e.g., the XHand), we install the same type of sensor as used in the hand. In our setup, this sensor is a magnet-based tactile array capable of measuring three-dimensional forces across 120 points on its surface. The force data is output via an SPI communication interface using a proprietary protocol. By configuring this interface on our embedded system, the force array can be successfully transmitted to the host machine.

# E  Data Collection and Policy Training details

## E.1  Data Collection

We collected 310 trajectories for Cube Picking task policy training, 175 trajectories for Egg Carton Opening task policy training, and 400 trajectories for Tea Picking Using Tools policy training (for both Inspire Hand and XHand). For the kitchen task, we collected 370 trajectories covering all four sub-tasks, plus an additional 100 trajectories focused solely on knob closing.

For the Inspire Hand, all data types—including wrist position from ARKit, policy visual observations from the wrist-mounted camera, joint angles from encoders, and tactile feedback—were recorded at 45 FPS. For the XHand, we recorded at 30 FPS, as the tactile sensor readings became unstable at higher recording frequencies. For each data type, we recorded the receive timestamp $t_{\text{receive}}$ when the data arrived at the recording buffer.

We wear green gloves when collecting data with exoskeleton as we use green PLA-CF to 3D-printed the exoskeleton. We found consistent color helps SAM2 to yield better segmentation results.

## E.2  Training Data Latency Management

There is an inherent latency between the time when sensors capture data and when that data actually arrives in the recording buffer. To ensure our imitation learning policy receives properly aligned observations (visual observations, tactile sensor readings) and actions (joint encoder readings), we calculate the actual data capture time using $t_{\text{capture}} = t_{\text{receive}} - l_{\text{sensor}}$, where $l_{\text{sensor}}$ refers to the latency from capture to receive for a particular sensor. We measure the iPhone and OAK camera latency by reading a rolling QR code displayed on a computer monitor showing the current computer system time, as proposed in UMI [16]. The camera and iPhone latency is calculated as $l_{\text{camera}} = t_{\text{receive}} - t_{\text{display}} - l_{\text{display}}$, where $l_{\text{display}}$ represents the monitor refresh rate.

The encoder latency is adjusted by examining the overlay image between the recorded exoskeleton image and the corresponding robot hand image from action replay. If the encoder latency is set

too high, the robot hand fingers will execute future actions and lead in the overlay image. If the encoder latency is set too low, the robot hand fingers will lag behind the exoskeleton fingers in the overlay image. We tune the encoder latency until the exoskeleton fingers and robot fingers are perfectly aligned. Once all data timestamps are adjusted, we linearly interpolate the joint angles and tactile readings to obtain data points properly aligned with the camera timestamps. Finally, We downsample the data by a factor of 3 to reduce the policy training time.

## E.3  Policy Training

We process the visual observations with pretrained DINO-V2 [17, 18]. Before passing the visual observations into DINO-V2, we augment it with random crop, color jitter, random grayscale and Gaussian Blur. We concatenate the CLS token from DINO-V2 with tactile sensor readings as input to the diffusion policy [19, 20]. The policy predicts 16 steps of robot actions, which contain both 6-DoF robot end-effector relative actions and hand actions (6-DoF for Inspire Hand and 12-DoF for XHand). We train the models for 400 epochs across all tasks for both types of hands. The pretrain DINO-V2 is not frozen and updated during the policy training.

