# OpenReview forum: "DexUMI: Using Human Hand as the Universal Manipulation Interface for Dexterous Manipulation"
_robot-learning.org/CoRL/2025/Workshop/Dexterous_Manipulation — CoRL 2025 Workshop Dexterous Manipulation Spotlight_

### Official Review · Reviewer_Ankp · 2025-09-12

**Rating:** 9
**Confidence:** 5

**Review:**

DexUMI introduces a new framework for collecting human demonstrations that can easily be converted to robots. They also showcase its applications on different manipulation tasks. The major limitation of this framework would be the potential scalability of it to newer hands but I think it's still a very strong paper that should be accepted.

---

### Decision · Program_Chairs · 2025-09-18

Accept (Spotlight)